molecular computing/theory of computing/hybrid computing

actin network, computing, waves, logical gates, finite-state machine, automata

**Author for correspondence:**
Andrew Adamatzky
e-mail: andrew.adamatzky@uwe.ac.uk

# Actin droplet machine

Andrew Adamatzky[1], Jörg Schnauß[2] and Florian Huber[3]

[1]Unconventional Computing Laboratory, Department of Computer Science, University of the West of England, Bristol, UK
[2]Soft Matter Physics Division, Peter Debye Institute for Soft Matter Physics, Faculty of Physics and Earth Sciences, Leipzig University, Germany & Fraunhofer Institute for Cell Therapy and Immunology (IZI), DNA Nanodevices Unit, Leipzig, Germany
[3]Netherlands eScience Center, Science Park 140, 1098 XG Amsterdam, The Netherlands

 AA, 0000-0003-1073-2662; JS, 0000-0002-6408-8676

The actin droplet machine is a computer model of a three-dimensional network of actin bundles developed in a droplet of a physiological solution, which implements mappings of sets of binary strings. The actin bundle network is conductive to travelling excitations, i.e. impulses. The machine is interfaced with an arbitrary selected set of $k$ electrodes through which stimuli, binary strings of length $k$ represented by impulses generated on the electrodes, are applied and responses are recorded. The responses are recorded in a form of impulses and then converted to binary strings. The machine's state is a binary string of length $k$: if there is an impulse recorded on the $i$th electrode, there is a '1' in the $i$th position of the string, and '0' otherwise. We present a design of the machine and analyse its state transition graphs. We envisage that actin droplet machines could form an elementary processor of future massive parallel computers made from biopolymers.

## 1. Introduction

Actin is a protein presented in forms of monomeric, globular actin (G-actin) and filamentous actin (F-actin) [1–3]. G-actin polymerizes into filamentous actin forming a double helical structure [4–6]. The filaments can be further arranged into bundles by various different mechanisms such as crowding effects, cross-linking or counter-ion condensation [7–15]. The bundles are conductive to travelling localizations—defects, ionic waves, solitons [16–25]. By interpreting the presence or absence of a travelling localization at a given site of the network at a given time step, we can implement logical functions. This approach was comprehensively developed and successfully tested on chemical systems in the framework of collision-based computing [26–31]. As actin networks can implement logical functions, they can compute. So, in [32] we proposed a road map to experimental implementation of cytoskeleton-based computing devices. We proposed that collision-based cytoskeleton computers implement logical gates via interactions between travelling localization: voltage solitons on actin filaments or tubulin

microtubules bundles. An architecture of cytoskeleton computers can be developed via programmable polymerization of actin networks. Such cytoskeleton computers would take data via electrical and optical means, the signals (solitons, conformational defects) initiated by the input stimuli will be travelling along the network and the computational will be implemented via collisions of the signals at the structural gates of the network.

Our approach—computing with excitation waves propagating on overall 'density' of the conductive material—has previously been presented by us in [33]. As conductive material we looked at networks of actin bundles which were arranged by crowding effects without the need of additional accessory proteins [10,11]. We demonstrated how to discover logical gates on a two-dimensional slice of the actin bundle network by representing Boolean inputs and outputs as spikes of the network activity. In a previous paper [33], we demonstrated, using numerical integration of FitzHugh–Nagumo model, that a two-dimensional actin network realized $k$-ary Boolean functions $G : \{0, 1\}^k \to \{0, 1\}$, when $k$ input electrodes and one output electrodes are employed.

In the present paper, we develop a novel concept and computer modelling implementation of the actin network machine, which implements a mapping $F : \{0, 1\}^k \to \{0, 1\}^k$, where $k$ is a number of electrodes, and '1' signifies the presence of an impulse on the electrode and '0' the absence. At a higher level, the machine acts as a finite-state machine, at the lower level, a structure of the mapping $F$ is determined by interactions of impulses propagating on the three-dimensional network of actin bundles.

We also offer an alternative to a numerical integration used in [33]: an automaton model of a three-dimensional actin network. There is a substantial body of evidence confirming that automaton models are sufficient and appropriate discrete tools for modelling dynamics of spatially extended nonlinear excitable media [34–36], propagation [37], action potential [38,39], electrical pulses in the heart [40–42]. A major advantage of automata is that they require less computational resources than typical numerical integration approaches.

Results presented in the paper give a rather 'computer engineering' view on a computation implementable with travelling localizations on acting bundle networks. We do not speculate about potential biological meanings of the phenomena described. That could be a scope of future studies.

The paper is structured as follows. Our modelling approach is described in detail in §2. This includes a representation of a three-dimensional actin bundle network (§2.1), a structure of an automaton model to simulate propagation of impulses on the actin bundle network (§2.2), and an interface with the actin network (§2.3). In §2.4, we analyse dependencies of a number of Boolean gates implemented in the network on an excitation threshold and refractory period. Thus, we justify the selection of these parameters for the construction of the actin machine. The actin droplet machine is designed and analysed in §3. Section 4 discusses the results in a context of cytoskeleton computing and outlines directions for future research.

# 2. Methods

The overall approach is the following: we simulate the actin bundle network using three-dimensional arrays of finite-state machines, cellular automata. We select several domains of the network and assign them as inputs and outputs. We represent Boolean logic values with spikes of electrical activity, which are schematically represented as a virtual experiment in figure 1. We stimulate the network with all possible configurations of input strings and record spikes on the outputs. Based on the mapping of configurations of input spikes to output spikes, we reconstruct logical functions implemented by the network. In our design of the actin droplet machine, we consider outputs recorded on all electrodes at a given time step as a binary string and then represent the actin droplet machine as a finite-state machine whose states are binary strings of a given length.

## 2.1. Three-dimensional actin network

As a template for our actin droplet machine, we used an actual three-dimensional actin bundle network produced in laboratory experiments with purified proteins (figure 2). The underlying experimental method was shown to reliably produce regularly spaced bundle networks from homogeneous filament solutions inside small isolated droplets in the absence of molecular motor-driven processes or other accessory proteins [15]. These structures effectively form very stable and long-living three-dimensional networks, which can be readily imaged with confocal microscopy resulting in stacks of optical two-dimensional slices (figure 2). Dimensions of the network are the following: size along $x$

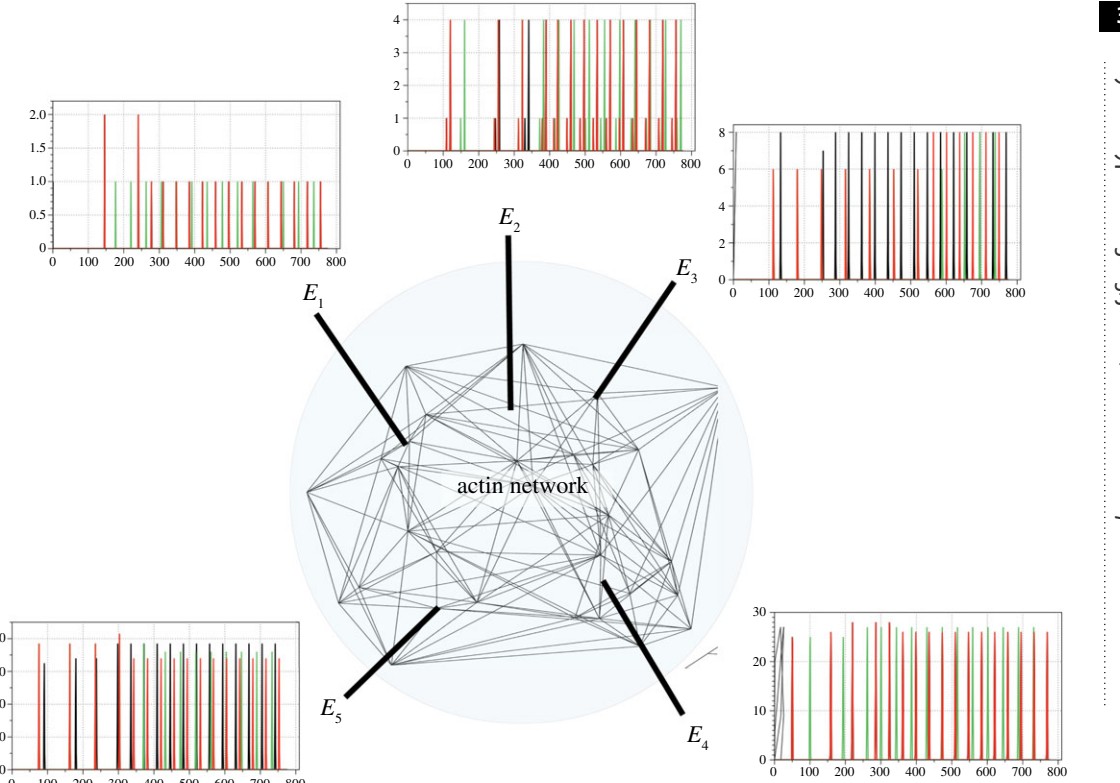

**Figure 1.** A scheme of a virtual experiment. The actin bundle network is shown as a three-dimensional Delaunay triangulation. Electrodes are shown by thick lines and labelled $E_1$ to $E_5$. Exemplary trains of spikes are shown near the electrodes.

coordinate is 225 µm (width), along $y$ coordinate is 222 µm (height), along $z$ coordinate is 112 µm (depth), voxel width is 0.22 µm, height 0.22 µm and depth 4 µm.

Original image: $A_z = (a_{ijz})_{1 \leq i,j \leq n, 1 \leq z \leq m}$, $a_{ijz} \in \{r_{ijz}, g_{ijz}, b_{ijz}\}$, where $n = 1024$, $m = 30$, $r_{ijz}$, $g_{ijz}$, $b_{ijz}$ are RGB values of the element at $ijz$, $1 \leq r_{ijz}$, $g_{ijz}$, $b_{ijz} \leq 255$ was converted to a conductive matrix $\mathbf{C} = (c_{ijz})_{1 \leq i,j \leq n, 1 \leq z \leq m}$ as follows: $c_{ijz} = 1$ if $r_{ijz} > 40$, $g_{ijz} > 19$ and $b_{ijz} > 19$. The conductive matrices are shown in figure 3. The three-dimensional conductive matrix is compressed along the $z$-axis to reduce consumption of computational resources, scenario of the non-compressed matrix will be considered in future papers.

## 2.2. Automaton model

To model activity of an actin bundle network we represent it as an automaton $\mathcal{A} = \langle \mathbf{C}, \mathbf{Q}, r, h, \theta, \delta \rangle$. $\mathbf{C} \subset \mathbf{Z}^3$ is a set of voxels, or a conductive matrix $\mathbf{C}$ defined in §2.1. Each voxel $p \in \mathbf{C}$ takes states from the set $\mathbf{Q} = \{\bigstar, \bullet, \circ\}$, excited ($\bigstar$), refractory ($\bullet$), resting ($\circ$) and is complemented by a counter $h_p$ to handle the temporal decay of the refractory state. Following discrete time steps, each voxel $p$ updates its state depending on its current state and the states of its neighbourhood $u(p) = \{q \in \mathbf{C} : d(p, q) \leq r\}$, where $d(p, q)$ is an Euclidean distance between voxels $p$ and $q$; $r \in \mathbf{N}$ is a neighbourhood radius. $\theta \in \mathbf{N}$ is an excitation threshold and $\delta \in \mathbf{N}$ is refractory delay. All voxels update their states in parallel and by the same rule:

$$p^{t+1} \begin{cases} \bigstar, & \text{if } (p^t = \circ) \text{ and } (\sigma(p)^t > \theta) \\ \bullet, & \text{if } (p^t = \bigstar) \text{ or } ((p^t = \bullet) \text{ and } (h_p^t > 0)) \\ \circ, & \text{otherwise} \end{cases}$$

and

$$h_p^{t+1} = \begin{cases} \delta, & \text{if } (p^{t+1} = \bullet) \text{ and } (p^t = \bigstar) \\ h_p^t - 1, & \text{if } (p^{t+1} = \bullet) \text{ and } (h_p^t > 0) \\ 0, & \text{otherwise.} \end{cases}$$

Every resting ($\circ$) voxel of $\mathbf{C}$ excites ($\bigstar$) at the moment $t + 1$ if a number of its excited neighbours at the moment $t$, $\sigma(p)^t = |\{q \in u(p) : q^t = \bigstar\}|$, exceeds a threshold $\theta$. An excited voxel $p^t = \bigstar$ takes the refractory

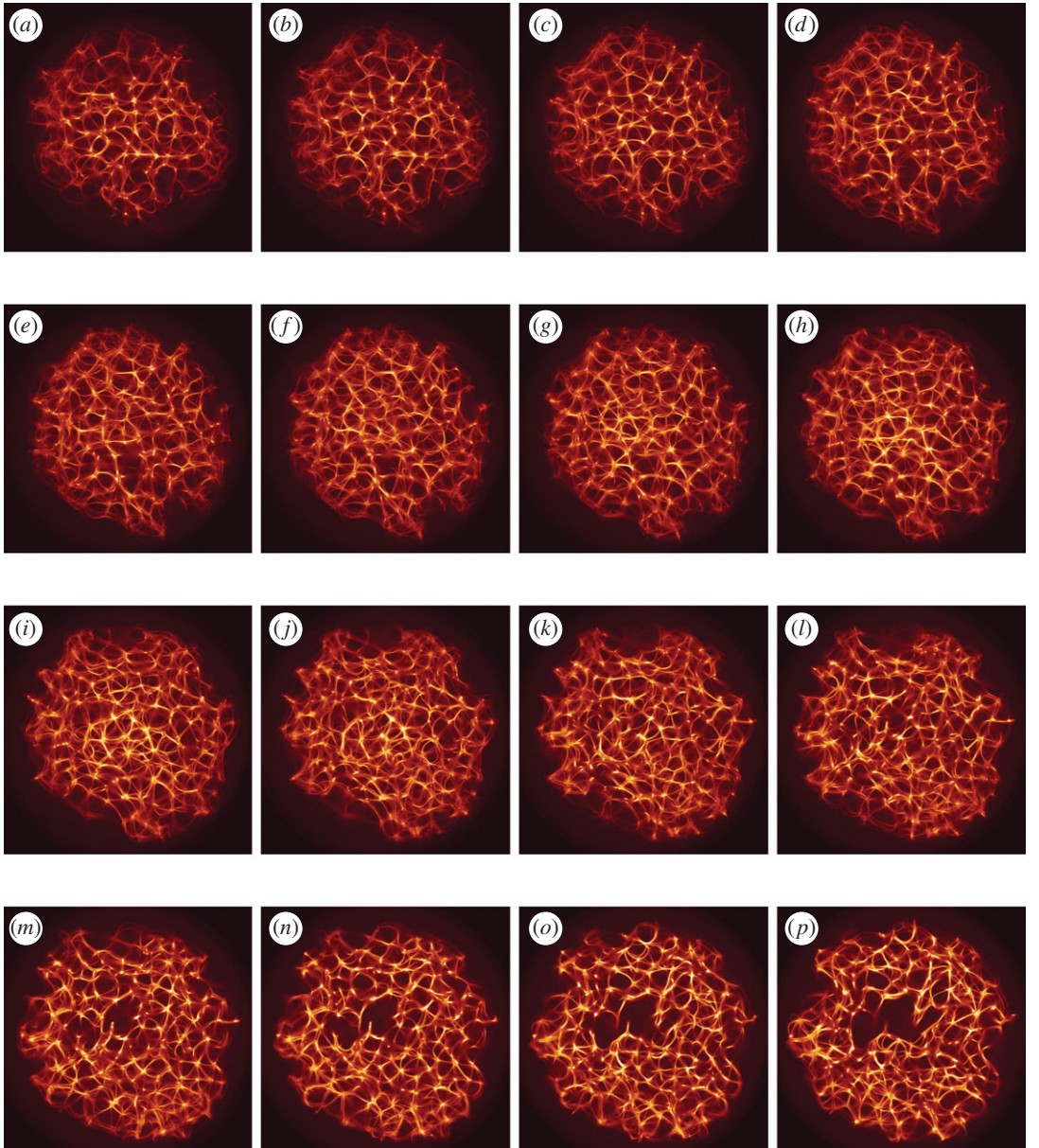

**Figure 2.** Exemplary $z$-slices of a three-dimensional actin bundle network reconstructed as described in [15]. (a) $z = 1$, (b) $z = 2$, (c) $z = 3$, (d) $z = 4$, (e) $z = 5$, (f) $z = 6$, (g) $z = 7$, (h) $z = 8$, (i) $z = 9$, (j) $z = 10$, (k) $z = 11$, (l) $z = 12$, (m) $z = 13$, (n) $z = 14$, (o) $z = 15$ and (p) $z = 16$.

state • at the next time step $t + 1$ and at the same moment a counter of refractory state $h_p$ is set to the refractory delay $\delta$. The counter is decremented, $h_p^{t+1} = h_p^t - 1$ at each iteration until it becomes 0. When the counter $h_p$ becomes zero the voxel $p$ returns to the resting state ∘. For all results shown in this manuscript, the neighbourhood radius was set to $r = 3$. Choices of $\theta$ and $\delta$ are considered in §2.4.

## 2.3. Interfacing with the network

To stimulate the network and to record activity of the network, we assigned several domains of **C** as electrodes. We calculated a potential $p_x^t$ at an electrode location $c \in \mathbf{C}$ as $p_c = |z : d(c, z) < r_e$ and $z^t = \star|$, where $d(c, z)$ is an Euclidean distance between sites $x$ and $z$ in three-dimensional space. We have chosen an electrode radius of $r_e = 4$ voxels and conducted two families of experiments with two configurations of electrodes.

In the first family of experiments $\mathcal{E}_1$, we studied frequencies of two-input-one-output Boolean functions implementable in the network. We used 10 electrodes, their coordinates are listed in table 1 and a configuration is shown in figure 4a. Electrodes $E_0$ representing input $x$ and $E_9$ representing input $y$ are

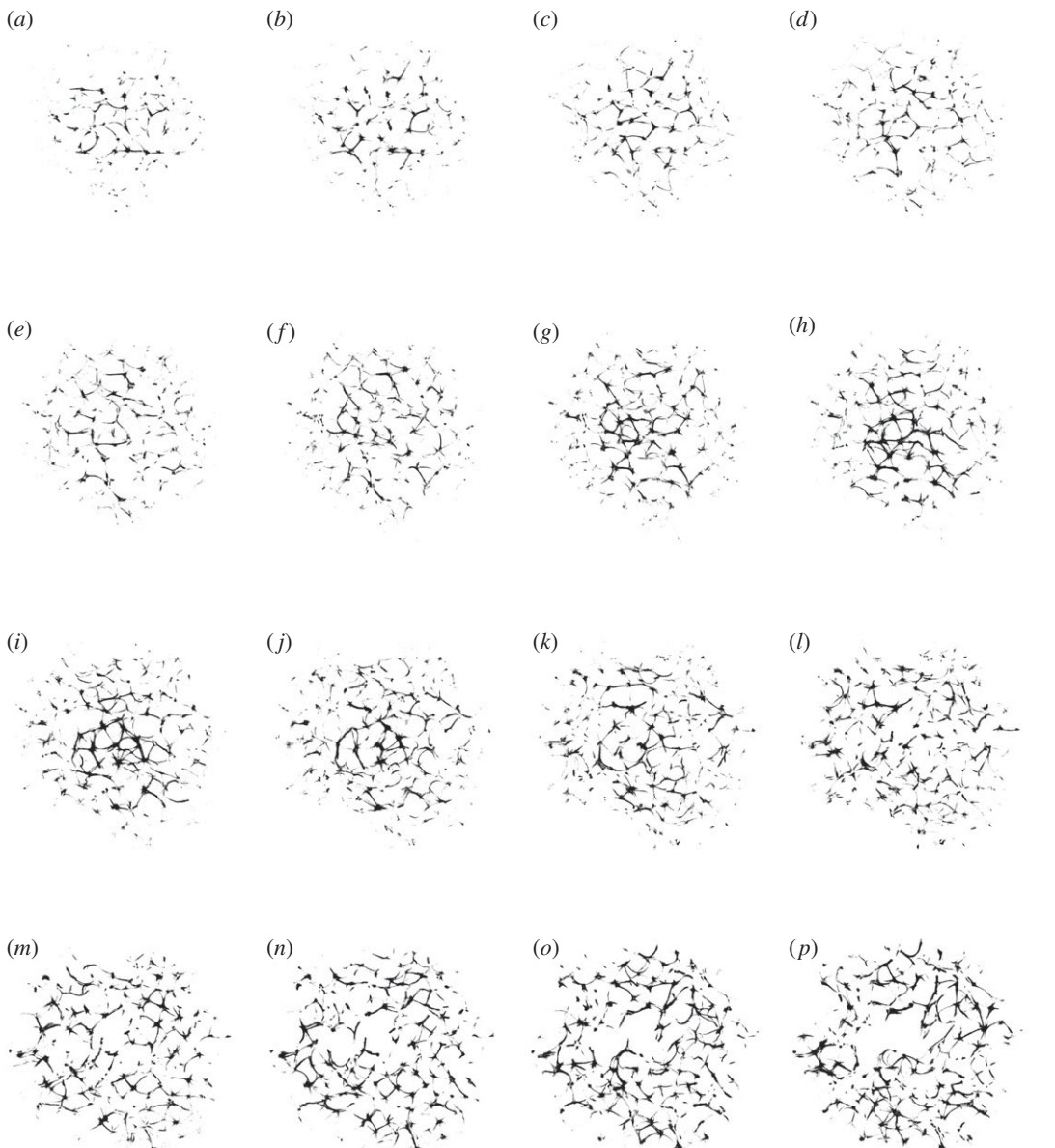

**Figure 3.** Exemplary z-slices of 'conductive' geometries C selected from the three-dimensional actin bundle network shown in figure 2, which were reconstructed as described in [15]. (a) $z = 1$, (b) $z = 2$, (c) $z = 3$, (d) $z = 4$, (e) $z = 5$, (f) $z = 6$, (g) $z = 7$, (h) $z = 8$, (i) $z = 9$, (j) $z = 10$, (k) $z = 11$, (l) $z = 12$, (m) $z = 13$, (n) $z = 14$, (o) $z = 15$ and (p) $z = 16$.

the input electrodes, all others are output electrodes representing outputs $z_1, \ldots, z_8$. Results are presented in §2.4. In the second family of experiments $\mathcal{E}_2$, we used six electrodes (table 2 and figure 4b). All electrodes were considered as inputs during stimulation and outputs during recording of the network activity.

Exemplary snapshots of excitation dynamics on the network are shown in figure 5. Domains corresponding to the two electrodes $e_0$ and $e_9$ (table 1 and figure 4a) have been excited (figure 5a). The excitation wave fronts propagates away from $e_0$ and $e_9$ (figure 5b). The fronts traverse the whole breadth of the network (figure 5c). Due to the presence of circular conductive paths in the network, the repetitive patterns of activity emerge (figure 5d). Recordings of potential and videos of experiments are available within the Zenodo repository [43].

## 2.4. Selecting excitation threshold and refractory delay to maximize a number of logical gates

To design an actin droplet machine with complex behaviour, we need to find values of refractory delay and excitation threshold for which the actin bundles network executes a maximum of Boolean gates. To map dynamics of the network onto sets of gates, we undertook the following trials of stimulation:

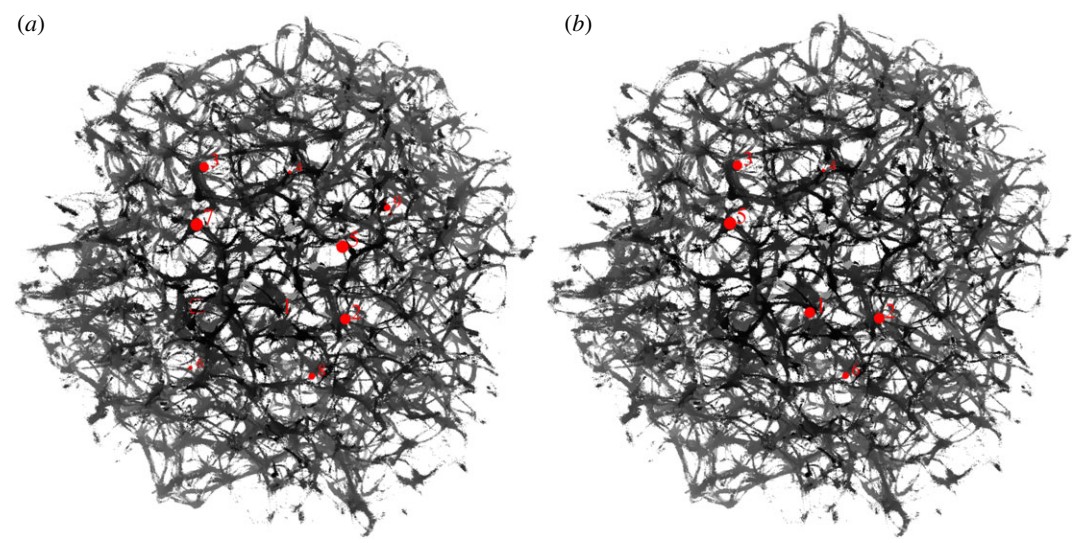

**Figure 4.** Configurations of electrodes in the three-dimensional network of actin bundles used in (a) $\mathcal{E}_1$ and (b) $\mathcal{E}_2$. Depth of the network is shown by level of grey. Sizes of the electrodes are shown in perspective.

**Table 1.** Coordinates of electrodes in experiments family $\mathcal{E}_1$.

| e | i | j | z |
|---|---|---|---|
| 1 | 369 | 567 | 6 |
| 2 | 509 | 580 | 10 |
| 3 | 631 | 590 | 10 |
| 4 | 382 | 322 | 12 |
| 5 | 533 | 331 | 23 |
| 6 | 626 | 463 | 7 |
| 7 | 358 | 676 | 22 |
| 8 | 369 | 424 | 7 |
| 9 | 572 | 691 | 17 |
| 10 | 705 | 394 | 17 |

1. fixed refractory delay $\delta = 20$ and excitation threshold $\theta = 4, 5, \ldots, 12$,
2. fixed excitation threshold $\theta = 7$, and refractory delay $\delta = 10, 15, 17, \ldots, 24, 30$.

An example of the network spiking activity as a response to stimulation is shown in figure 1. We stimulated the network with all possible configurations of inputs, recorded the network's electrical dynamics and then extracted logical gates as follows. For each possible combination $(i, j, k)$, $1 \leq i, j, k \leq 6$, $i \neq j$, $i \neq k$, $j \neq k$, we considered electrodes $\mathcal{E}_i$ and $\mathcal{E}_j$ to be inputs, representing Boolean variables $x$ and $y$, respectively, and electrode $\mathcal{E}_k$ as output electrode, representing the result of a Boolean function. To input $x =$ TRUE, we applied a current to electrode $\mathcal{E}_i$, to input $y =$ TRUE to electrode $\mathcal{E}_j$. Then we recorded the potential at electrode $\mathcal{E}_k$. Two-input-one-output logical functions were extracted from the spiking events as follows. Assume each spike represents logical TRUE and that spikes being less than six iterations closer to each other happen at the same moment. Then a representation of gates by spikes and their combination will be as shown in figure 6.

For each combination $(\rho, \theta)$, we counted the numbers of gates OR $(x + y)$, AND $(xy)$, XOR $(x \oplus)$, NOT-AND $(\overline{x}y)$, AND-NOT $(x\overline{y})$ and SELECT $(x, y)$. We found that overall the total number of gates $v(\theta)$ realized by the network decreases with increase of $\theta$ (figure 7a). The function $v(\theta)$ is nonlinear and could be adequately described by a five degree polynomial. The function reaches its maximal value at $\theta = 7$ (figure 7a). OR gates are most commonly realized at $\theta = 11$, AND gates at $\theta = 6$ and *xor* gates at $\theta = 5$ as well as $\theta = 7$

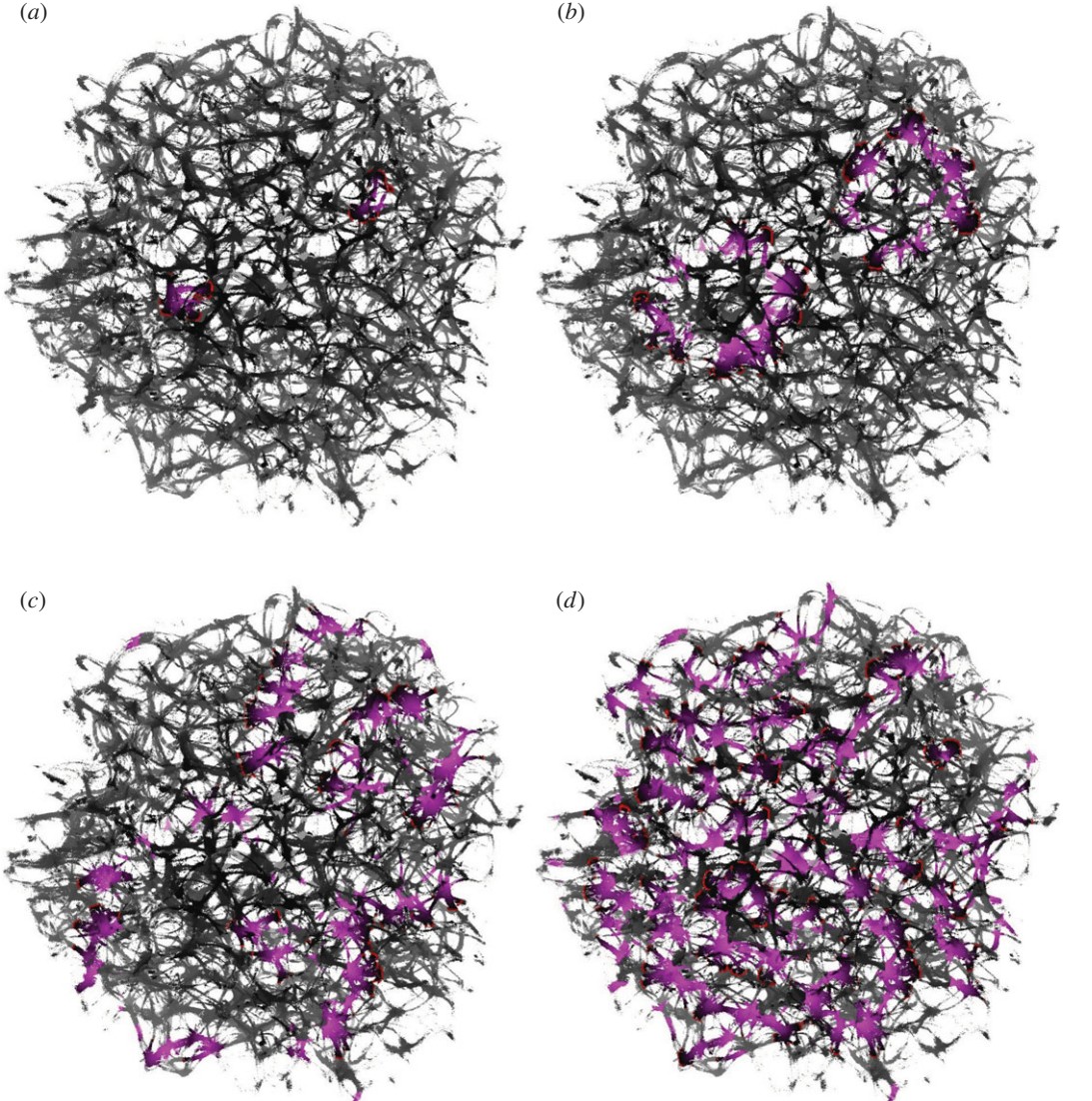

**Figure 5.** Snapshots of excitation dynamics on the network. The excitation wavefront is red and the refractory tail is magenta. The excitation threshold is $\theta = 7$ and the refractory delay is $\delta = 20$, (a) $t = 13$, (b) $t = 50$, (c) $t = 200$ and (d) $t = 500$.

**Table 2.** Coordinates of electrodes in experiments family $\mathcal{E}_2$.

| e | i | j | z |
|---|---|---|---|
| 1 | 369 | 567 | 6 |
| 2 | 509 | 580 | 10 |
| 3 | 631 | 590 | 10 |
| 4 | 382 | 322 | 12 |
| 5 | 533 | 331 | 23 |
| 6 | 369 | 424 | 7 |
| 7 | 572 | 691 | 17 |
| 8 | 705 | 394 | 17 |

(figure 7b). A number of AND-NOT gates implemented by the network reaches its highest value at $\theta = 6$ then drops sharply after $\theta_8$ (figure 7c). NOT-AND gates are more common at $\theta = 5, 7, 9, 11$, while SELECT($x$) has its peak at $\theta = 7$ and SELECT($y$) at $\theta = 8, 9$ (figure 7c). A total number of gates realized in the network with the excitability threshold fixed to $\theta = 7$ decreases with the increase of $\delta$. Oscillations

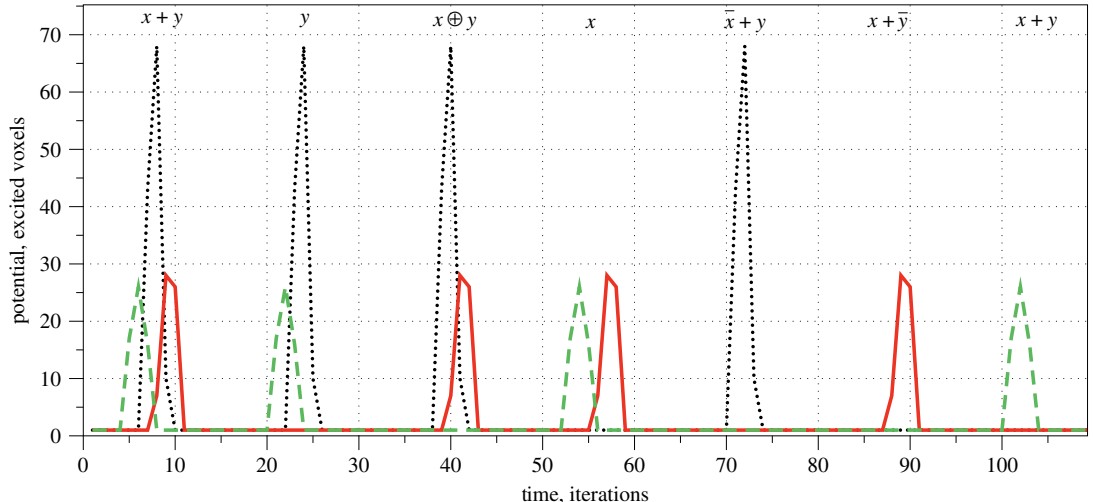

**Figure 6.** Representation of two-inputs-one-output Boolean gates by combinations of spikes. Black dotted line shows the potential at an output electrode when the network was stimulated by input pair $(x, y)$ =(F<small>ALSE</small>, T<small>RUE</small>), red solid by (T<small>RUE</small>, F<small>ALSE</small>) and green dashed by $(x, y)$ =(T<small>RUE</small>, T<small>RUE</small>).

of $v(\delta)$ are visible at $15 \leq \delta \leq 25$ (figure 7$d$). The three highest values of $v(\delta)$ are achieved at $\delta = 10$, 17 and 20. Let us look now at the dependence of the numbers of OR, AND and XOR gates of the refractory delay $\delta$ in figure 7$e$. The number of OR gates increases with $\delta$ increasing from 10 to 15, but then drops substantially at $\delta = 18$ to reach its maximum at $\delta = 19$. Numbers of gates AND and XOR behave similarly to each other. They both have a pronounced peak at $\delta = 20$ (figure 7$e$).

The gate frequency analysis presented in this section allows us to choose $\theta = 7$ and $\delta = 20$ for an actin droplet machine constructed in the next section.

# 3. Actin droplet machine

An actin droplet machine is defined as a tuple $\mathcal{M} = \langle \mathcal{A}, k, \mathbf{E}, \mathbf{S}, F \rangle$, where $\mathcal{A}$ is an actin network automaton, defined in §2.2, $k$ is a number of electrodes, $\mathbf{E}$ is a configuration of electrodes, $\mathbf{S} = \{0, 1\}^k$, $F$ is a state-transition function $F : \mathbf{S} \to \mathbf{S}$ that implements a mapping between sets of all possible configurations of binary strings of length $k$. In the experiments reported here $k = 6$.

In our experiments, we have chosen six electrodes, their locations are shown in figure 4$b$ and exact coordinates in table 2. Thus, $F : \{0, 1\}^6 \to \{0, 1\}^6$ and the machine $\mathcal{M}$ has 64 states. We represent the inputs and the machine states in decimal encoding. Spikes detected in response to every input from $\{0, 1\}^6$ are shown in figure 8.

Global transition graphs of $\mathcal{M}$ for selected inputs are shown in figure 9. Nodes of the graphs are states of $\mathcal{M}$, edges show transitions between the states. These directed graphs are defined as follows. There is an edge from node $a$ to node $b$ if there is such $1 \leq t \leq 1000$ that $\mathcal{M}^t = a$ and $\mathcal{M}^{t+1} = b$.

Let us now define a weighted global transition graph $\mathcal{G} = \langle \mathbf{Q}, \mathbf{E}, w \rangle$, where $\mathbf{Q}$ is a set of nodes (isomorphic to the $\{0, 1\}^6$), and $\mathbf{E}$ is a set of edges, and weighting function $w : \mathbf{E} \to [0, 1]$ assigning a number of a unit interval to each edge. Let $a$, $b \in \mathbf{Q}$ and $e(a, b) \in \mathbf{E}$ then a normalized weight is calculated as $w(e(a, b)) = \left( \sum_{i \in \mathbf{Q}, t \in \mathbf{T}} \chi(s^t = a \text{ and } s^{t+1} = b) \right) \Big/ \left( \sum_{d \in \mathbf{Q}, t \in \mathbf{T}} \sum_{\mathbf{Q}, t \in \mathbf{T}} \chi(s^t = a \text{ and } s^{t+1} = d) \right)$, with $\chi$ takes value '1' when the conditions are true and '0' otherwise. In words, $w(e(a, b))$ is a number of transitions from $a$ to $b$ observed in the evolution of $\mathcal{M}$ for all possible inputs from $\mathcal{Q}$ during time interval $\mathbf{T}$ normalized by a total number of transition from $a$ to all other nodes. The graph $\mathcal{G}$ is visualized in figure 10$a$. Nodes which have predecessors are 1–6, 8–10, 12, 16–21, 24, 25, 28, 32–34, 36–38, 40, 41, 44, 48–50, 52, 53, 56. Nodes without predecessors are 7, 11, 13–15, 22, 23, 26, 27, 29–31, 35, 39, 42, 43, 45–47, 51, 54, 55, 57–63.

Let us convert $\mathcal{G}$ to an acyclic non-weighted graph of more likely transitions $\mathcal{G}^* \langle \mathbf{Q}, \mathbf{E}^* \rangle$, where $e(a, b) \in \mathbf{E}^*$ if $w(e(a, b)) = max\{w(e(a, c)) \mid e(a, c) \in \mathbf{E}\}$. That is for each node we select an outgoing edge with maximum weight. The graph is a tree, see figure 10$b$. Most states apart from 1, 2, 4, 8, 16, 20, 32 are Garden-of-Eden configurations, which have no predecessors. Indegrees $v()$ of not-Garden-of-Eden nodes are $v(20) = 1$,

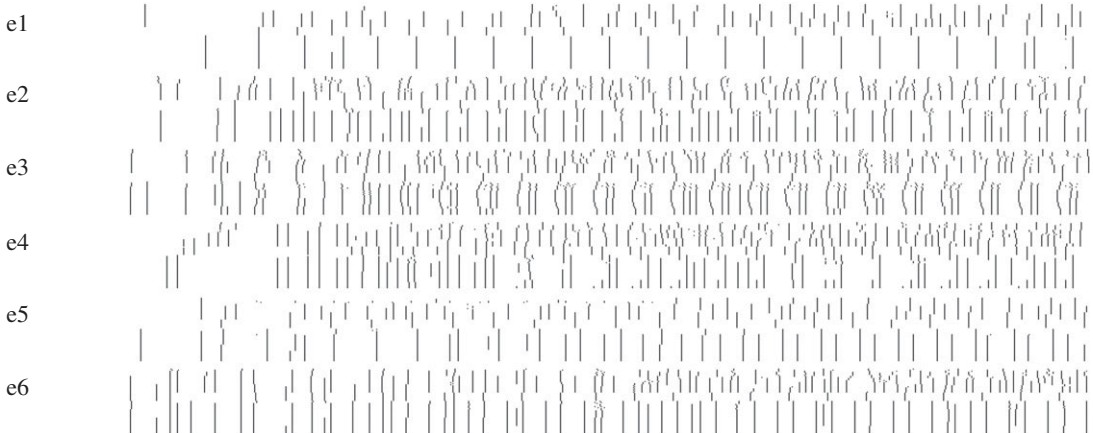

**Figure 7.** An average number $\nu$ of gates realizable on each of the electrodes $e_1$, ..., $e_8$ depends on threshold $\theta$ of excitation when the refractory delay $\delta$ is fixed to 20 (abc) and on refractory delays $\delta$ when the threshold $\theta$ is fixed to 7 (def). (a) Number of gates $\nu$ versus threshold $\theta$, $\delta = 20$. (b) Number of OR (black circle), AND (orange solid triangle) and XOR (red blank triangle) gates, $\delta = 20$. (c) Number of NOT-AND (yellow blank triangle), AND-NOT (magenta solid triangle), SELECT($x$) (cyan blank rhombus), SELECT($y$) (light blue disc), $\delta = 20$. (d) Number of gates $\nu$ versus delay $\delta$, $\theta = 7$. (e) Number of OR (black circle), AND (orange solid triangle) and XOR (red blank triangle) gates, $\theta = 7$.

**Figure 8.** All spikes recorded at each electrode for input binary strings from 1 to 63. The representation is implemented as follows. We stimulate the $\mathcal{M}$ with strings from $\{0, 1\}^6$ and represent a spike detected at time $t$ by a black pixel at position $t$ along horizontal axis. A plot of each electrode $e_i$ represents a binary matrix $\mathbf{S} = (s_{zt})$, where $1 \leq z \leq 63$ and $1 \leq t \leq 1000$: $s_{zt} = 1$ if the input configuration was $z$ and a spike was detected at moment $t$, and $s_{zt} = 1$ otherwise.

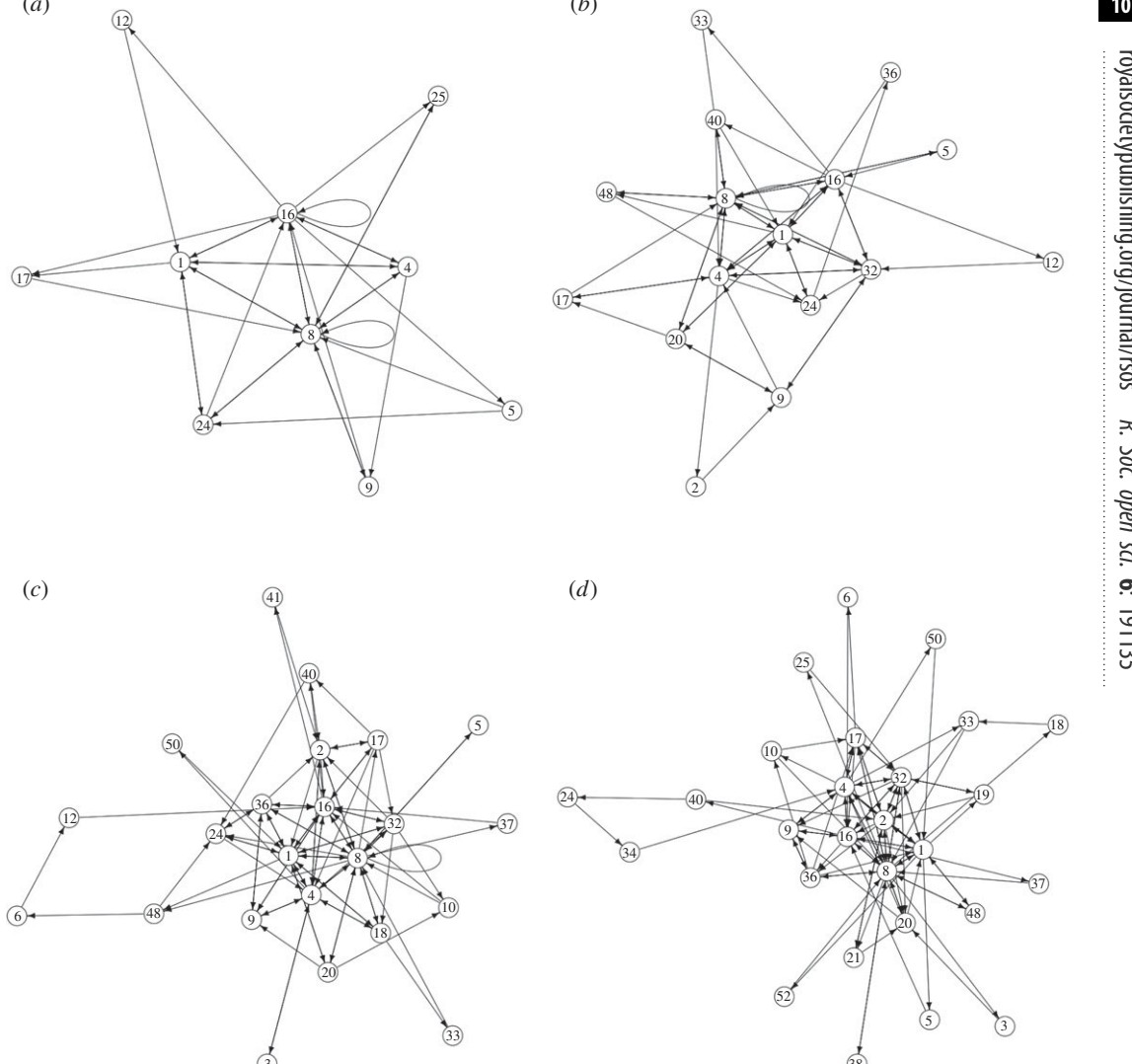

**Figure 9.** State transitions of machine $\mathcal{M}$ for selected inputs $l$. A node is a decimal encoding of the $\mathcal{M}$ state $(e_0^t \ldots e_5^t)$. (a) $l = 5$, (b) $l = 15$, (c) $l = 31$ and (d) $l = 63$.

$v(32) = 2$, $v(2) = 3$, $v(4) = 4$, $v(1) = 5$, $v(16) = 6$, $v(8) = 12$. There is one fixed point, the state 1, corresponding to the situation when a spike is recorded only on electrode $e_5$; it has no successors.

By analysing $\mathcal{G}$ we can characterize a richness of $\mathcal{M}$'s responses to input stimuli. We define a richness as a number of different states over all inputs, as shown in table 3, and distribution in figure 11a. A number of states produced increases from under five for beginning of $\mathcal{M}$ evolution and then reaches circa seven states on average. Oscillations around this value are seen in (figure 11a). Figure 11b shows a number of different nodes, generated in evolution of $\mathcal{M}$, stimulated by a given input. There are below 15 different states found in the evolution in responses to inputs 1 to 21 (21 corresponds to binary input string 010101); then a number of different nodes stay around 25. The diagram figure 11c shows how many inputs might lead to a given state/node of $\mathcal{M}$. Some of the states/nodes are seen to be Garden-of-Eden configurations **E** (nodes without predecessors) and thus could not be generated by stimulating $\mathcal{M}$ by sequences from **Q** − **E**.

Assume **T** is a set of temporal moments when the machine responded at least to one input string with a non-zero state. Configurations at each transition $t$ can be considered as outputs representing the function $g : \{0,1\}^6 \rightarrow \{0,1\}^6$. As we can see in table 3, transitions at $t = 41$ and $t = 53$ correspond to the highest number of different binary strings $(e_1, \ldots, e_6)$. The graph corresponding to $g(41)$ at $t = 41$ is shown in figure 12 and is not connected. The small component consists of fixed point 40 (string '101000') with two leafs 39 ('100111') and 38 ('100110'). The largest component has a tree structure at

(a)

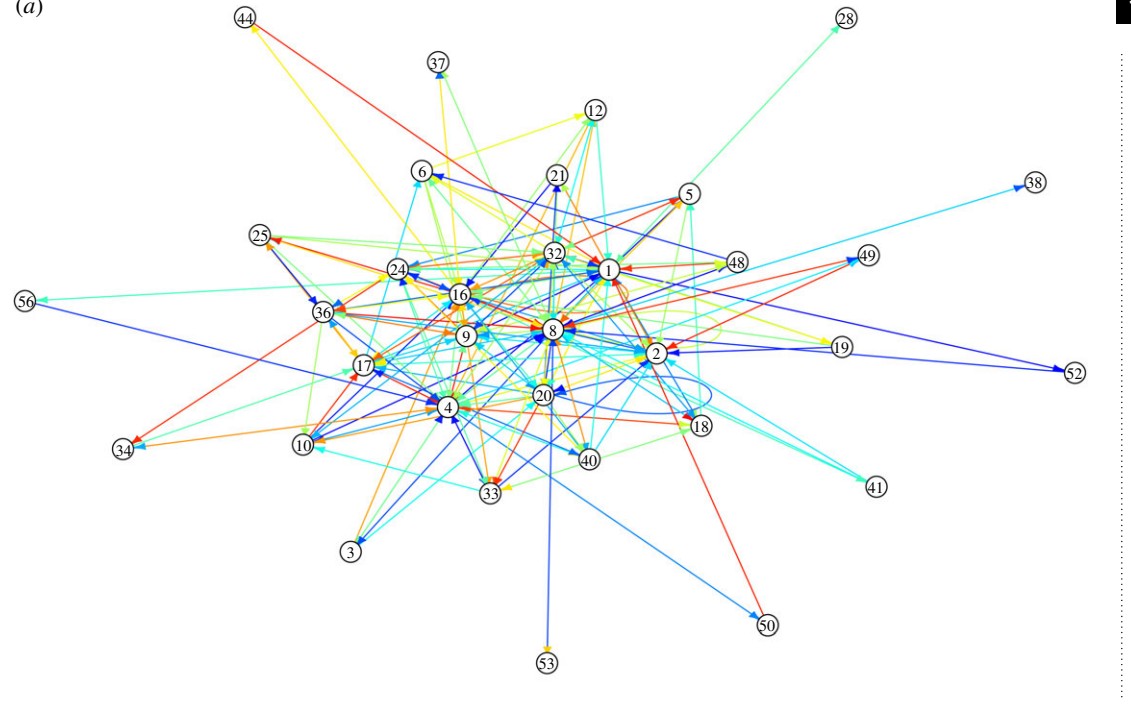

(b)

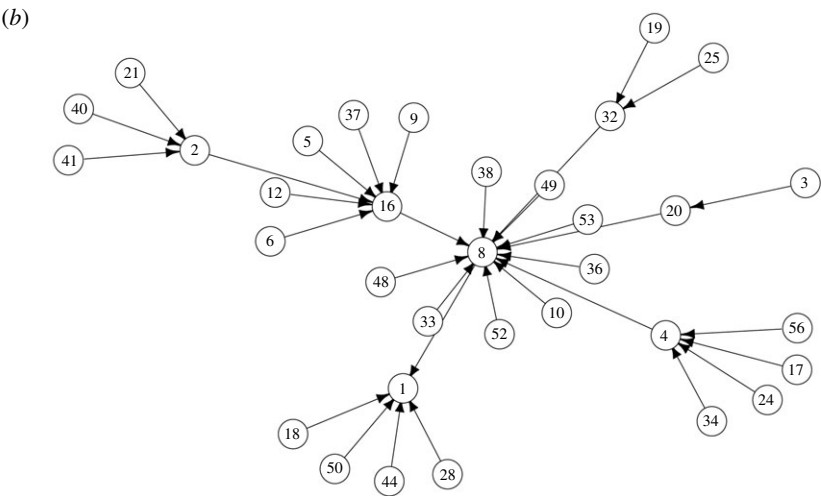

**Figure 10.** (a) Global graph of $\mathcal{M}$ state transitions. Edge weights are visualized by colours: from lowest weight in orange to highest weight in blue. (b) Pruned global graph of $\mathcal{M}$: only transitions with maximum weight for any given predecessors are shown, each node/state has at most one outgoing edge.

large, with cycle 2 ('000010')–1 ('000001') as a root. Other nodes with most predecessors are 8 ('001000'), 16 ('010000'), and 18 ('010010').

From the transitions $g(41)$, we can reconstruct Boolean functions realized at each of six electrodes (the functions are minimized and represented in a disjunctive normal form):

$$e_0 : f_0(x_0, \ldots, x_5) = x_0 \cdot \overline{x_1} \cdot x_2 \cdot x_3 + \overline{x_0} \cdot x_1 \cdot \overline{x_3} \cdot \overline{x_4} \cdot \overline{x_5} + x_0 \cdot \overline{x_1} \cdot x_2 \cdot \overline{x_3} \cdot x_4 + \overline{x_0} \cdot \overline{x_2} \cdot x_3 \cdot x_4 \cdot x_5 +$$
$$\overline{x_0} \cdot x_1 \cdot \overline{x_2} \cdot x_3 \cdot \overline{x_4} + \overline{x_1} \cdot x_2 \cdot \overline{x_3} \cdot \overline{x_4} \cdot x_5 + \overline{x_0} \cdot \overline{x_1} \cdot \overline{x_2} \cdot \overline{x_3} \cdot \overline{x_4} \cdot x_5 + \overline{x_0} \cdot x_1 \cdot \overline{x_2} \cdot x_3 \cdot x_4 \cdot \overline{x_5}$$

$$e_1 : f_1(x_0, \ldots, x_5) = x_0 \cdot \overline{x_1} \cdot x_2 \cdot x_3 + \overline{x_0} \cdot x_1 \cdot \overline{x_3} \cdot \overline{x_4} \cdot \overline{x_5} + x_0 \cdot \overline{x_1} \cdot x_2 \cdot \overline{x_3} \cdot x_4 + \overline{x_0} \cdot \overline{x_2} \cdot x_3 \cdot x_4 \cdot x_5 +$$
$$\overline{x_0} \cdot x_1 \cdot \overline{x_2} \cdot x_3 \cdot \overline{x_4} + \overline{x_1} \cdot x_2 \cdot \overline{x_3} \cdot \overline{x_4} \cdot x_5 + \overline{x_0} \cdot \overline{x_1} \cdot \overline{x_2} \cdot \overline{x_3} \cdot \overline{x_4} \cdot x_5 + \overline{x_0} \cdot x_1 \cdot \overline{x_2} \cdot x_3 \cdot x_4 \cdot \overline{x_5}$$

$$e_2 : f_2(x_0, \ldots, x_5) = x_0 \cdot \overline{x_1} \cdot x_2 \cdot x_3 + \overline{x_0} \cdot x_1 \cdot \overline{x_3} \cdot \overline{x_4} \cdot \overline{x_5} + x_0 \cdot \overline{x_1} \cdot x_2 \cdot \overline{x_3} \cdot x_4 + \overline{x_0} \cdot \overline{x_2} \cdot x_3 \cdot x_4 \cdot x_5 +$$
$$\overline{x_0} \cdot x_1 \cdot \overline{x_2} \cdot x_3 \cdot \overline{x_4} + \overline{x_1} \cdot x_2 \cdot \overline{x_3} \cdot \overline{x_4} \cdot x_5 + \overline{x_0} \cdot \overline{x_1} \cdot \overline{x_2} \cdot \overline{x_3} \cdot \overline{x_4} \cdot x_5 + \overline{x_0} \cdot x_1 \cdot \overline{x_2} \cdot x_3 \cdot x_4 \cdot \overline{x_5}$$

$$e_3 : f_3(x_0, \ldots, x_5) = x_0 \cdot \overline{x_1} \cdot x_2 \cdot x_3 + \overline{x_0} \cdot x_1 \cdot \overline{x_3} \cdot \overline{x_4} \cdot \overline{x_5} + x_0 \cdot \overline{x_1} \cdot x_2 \cdot \overline{x_3} \cdot x_4 + \overline{x_0} \cdot \overline{x_2} \cdot x_3 \cdot x_4 \cdot x_5 +$$
$$\overline{x_0} \cdot x_1 \cdot \overline{x_2} \cdot x_3 \cdot \overline{x_4} + \overline{x_1} \cdot x_2 \cdot \overline{x_3} \cdot \overline{x_4} \cdot x_5 + \overline{x_0} \cdot \overline{x_1} \cdot \overline{x_2} \cdot \overline{x_3} \cdot \overline{x_4} \cdot x_5 + \overline{x_0} \cdot x_1 \cdot \overline{x_2} \cdot x_3 \cdot x_4 \cdot \overline{x_5}$$

**Table 3.** Fifty-four state transitions of $\mathcal{M}$ over all possible inputs: $t$ is a transition step, $\mu(t)$ is a number of different states appeared over all possible inputs, $\mathbf{P}(t)$ is a set of nodes appeared at $t$.

| $t$ | $\mu(t)$ | $\mathbf{P}(t)$ |
|---|---|---|
| 1 | 3 | 8, 9, 1, |
| 2 | 3 | 16, 32, 8, |
| 3 | 3 | 1, 16, 32, |
| 4 | 3 | 8, 1, 16, |
| 5 | 3 | 1, 8, 16, |
| 6 | 3 | 16, 8, 1, |
| 7 | 4 | 8, 1, 16, 4, |
| 8 | 4 | 1, 16, 8, 5, |
| 9 | 5 | 16, 1, 8, 4, 5, |
| 10 | 4 | 16, 1, 8, 4, |
| 11 | 5 | 8, 1, 16, 20, 4, |
| 12 | 4 | 1, 16, 8, 20, |
| 13 | 6 | 16, 8, 1, 17, 4, 20, |
| 14 | 8 | 8, 16, 17, 4, 20, 1, 32, 2, |
| 15 | 8 | 1, 16, 8, 4, 2, 10, 20, 32, |
| 16 | 6 | 16, 4, 8, 1, 10, 32, |
| 17 | 5 | 16, 1, 4, 8, 9, |
| 18 | 7 | 8, 16, 4, 1, 17, 10, 9, |
| 19 | 6 | 1, 8, 16, 17, 4, 10, |
| 20 | 8 | 16, 1, 8, 17, 4, 24, 10, 2, |
| 21 | 9 | 8, 16, 1, 17, 32, 24, 9, 4, 10, |
| 22 | 6 | 16, 1, 8, 32, 9, 4, |
| 23 | 7 | 8, 1, 16, 4, 32, 9, 17, |
| 24 | 6 | 1, 16, 17, 4, 32, 8, |
| 25 | 7 | 16, 1, 8, 4, 17, 32, 9, |
| 26 | 6 | 8, 16, 4, 12, 1, 17, |
| 27 | 6 | 1, 8, 16, 4, 17, 32, |
| 28 | 6 | 16, 8, 4, 1, 24, 32, |
| 29 | 7 | 8, 1, 4, 16, 12, 24, 32, |
| 30 | 7 | 16, 1, 8, 4, 17, 2, 32, |
| 31 | 9 | 8, 1, 24, 16, 12, 4, 2, 17, 32, |
| 32 | 7 | 1, 16, 8, 24, 17, 2, 40, |
| 33 | 9 | 16, 8, 1, 4, 40, 17, 24, 32, 2, |
| 34 | 7 | 8, 1, 16, 24, 40, 4, 32, |
| 35 | 6 | 1, 16, 8, 4, 24, 2, |
| 36 | 6 | 16, 8, 1, 17, 4, 32, |
| 37 | 7 | 8, 16, 17, 4, 1, 40, 2, |
| 38 | 7 | 1, 8, 16, 17, 4, 24, 2, |
| 39 | 7 | 16, 1, 8, 17, 9, 4, 2, |
| 40 | 7 | 8, 16, 4, 1, 24, 40, 2, |
| 41 | 10 | 1, 8, 16, 9, 17, 4, 18, 24, 40, 2, |

(Continued.)

| $t$ | $\mu(t)$ | $\mathbf{P}(t)$ |
|---|---|---|
| 42 | 8 | 16, 1, 8, 4, 18, 33, 40, 24, |
| 43 | 9 | 8, 1, 16, 4, 24, 33, 18, 32, 34, |
| 44 | 9 | 1, 16, 8, 4, 17, 33, 24, 32, 40, |
| 45 | 7 | 16, 8, 4, 1, 12, 24, 34, |
| 46 | 7 | 8, 1, 16, 4, 24, 18, 34, |
| 47 | 5 | 1, 16, 8, 4, 33, |
| 48 | 5 | 16, 8, 1, 4, 17, |
| 49 | 8 | 8, 1, 16, 4, 20, 32, 24, 19, |
| 50 | 6 | 1, 16, 8, 4, 17, 32, |
| 51 | 8 | 16, 8, 1, 4, 17, 32, 41, 19, |
| 52 | 9 | 8, 16, 4, 1, 32, 33, 41, 2, 19, |
| 53 | 10 | 1, 8, 16, 4, 20, 10, 2, 41, 32, 19, |
| 54 | 9 | 16, 1, 8, 5, 17, 4, 2, 32, 19, |

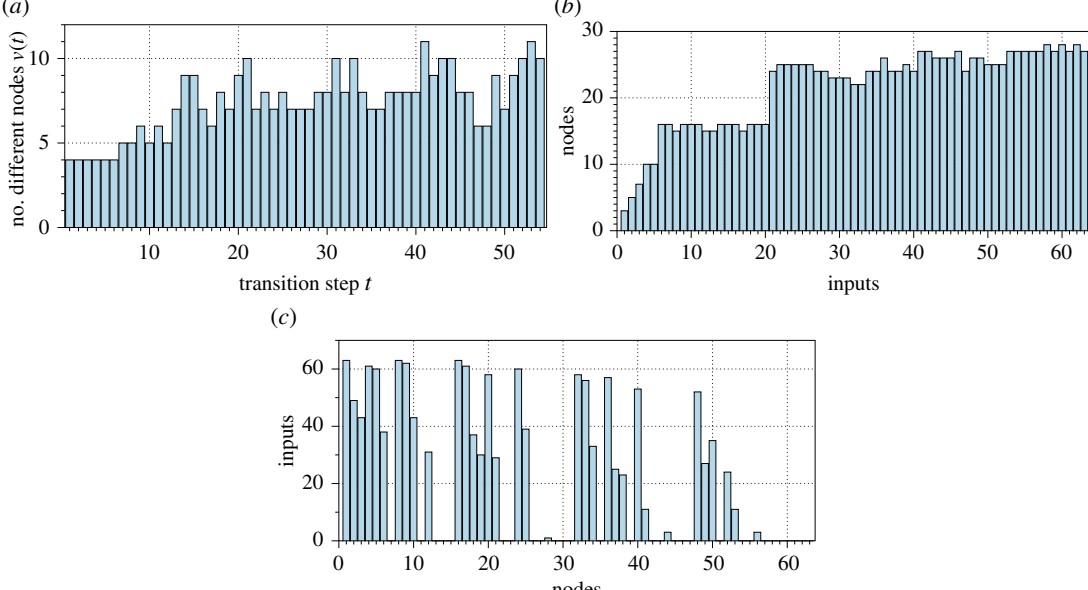

**Figure 11.** Distributions characterizing richness of $\mathcal{M}$'s responses. (*a*) Different states per transitions over all inputs. Horizontal axis shows steps of $\mathcal{M}$ transitions. Vertical axis is a number of different states. (*b*) Nodes per input. Horizontal axis shows decimal values of input strings. Horizon axis shows a number of different states/nodes generates in the evolution of $\mathcal{M}$. (*c*) Inputs per node.

$\text{e}_4 : f_4(x_0, \ldots, x_5) = x_0 \cdot \overline{x_1} \cdot x_2 \cdot x_3 + \overline{x_0} \cdot x_1 \cdot \overline{x_3} \cdot \overline{x_4} \cdot \overline{x_5} + x_0 \cdot \overline{x_1} \cdot x_2 \cdot \overline{x_3} \cdot x_4 + \overline{x_0} \cdot \overline{x_2} \cdot x_3 \cdot x_4 \cdot x_5 +$
$\quad \overline{x_0} \cdot x_1 \cdot \overline{x_2} \cdot x_3 \cdot \overline{x_4} + \overline{x_1} \cdot x_2 \cdot \overline{x_3} \cdot \overline{x_4} \cdot x_5 + \overline{x_0} \cdot \overline{x_1} \cdot \overline{x_2} \cdot \overline{x_3} \cdot \overline{x_4} \cdot x_5 + \overline{x_0} \cdot x_1 \cdot \overline{x_2} \cdot x_3 \cdot x_4 \cdot \overline{x_5}$

$\text{e}_5 : f_5(x_0, \ldots, x_5) = x_0 \cdot \overline{x_1} \cdot x_2 \cdot x_3 + \overline{x_0} \cdot x_1 \cdot \overline{x_3} \cdot \overline{x_4} \cdot \overline{x_5} + x_0 \cdot \overline{x_1} \cdot x_2 \cdot \overline{x_3} \cdot x_4 + \overline{x_0} \cdot \overline{x_2} \cdot x_3 \cdot x_4 \cdot x_5 +$
$\quad \overline{x_0} \cdot x_1 \cdot \overline{x_2} \cdot x_3 \cdot \overline{x_4} + \overline{x_1} \cdot x_2 \cdot \overline{x_3} \cdot \overline{x_4} \cdot x_5 + \overline{x_0} \cdot \overline{x_1} \cdot \overline{x_2} \cdot \overline{x_3} \cdot \overline{x_4} \cdot x_5 + \overline{x_0} \cdot x_1 \cdot \overline{x_2} \cdot x_3 \cdot x_4 \cdot \overline{x_5}.$

# 4. Discussion

Early concepts of sub-cellular computing on cytoskeleton networks as microtubule automata [44–46] and information processing in actin-tubulin networks [47] did not specify what type of 'computation' or 'information processing' the cytoskeleton networks could execute and how exactly they do this. We implemented several concrete implementations of logical gates and functions on a single actin filament [48] and on an intersection of several actin filaments [49] via collisions between solitons. We also used a reservoir-computing-like approach to discover functions on a single actin unit [50] and

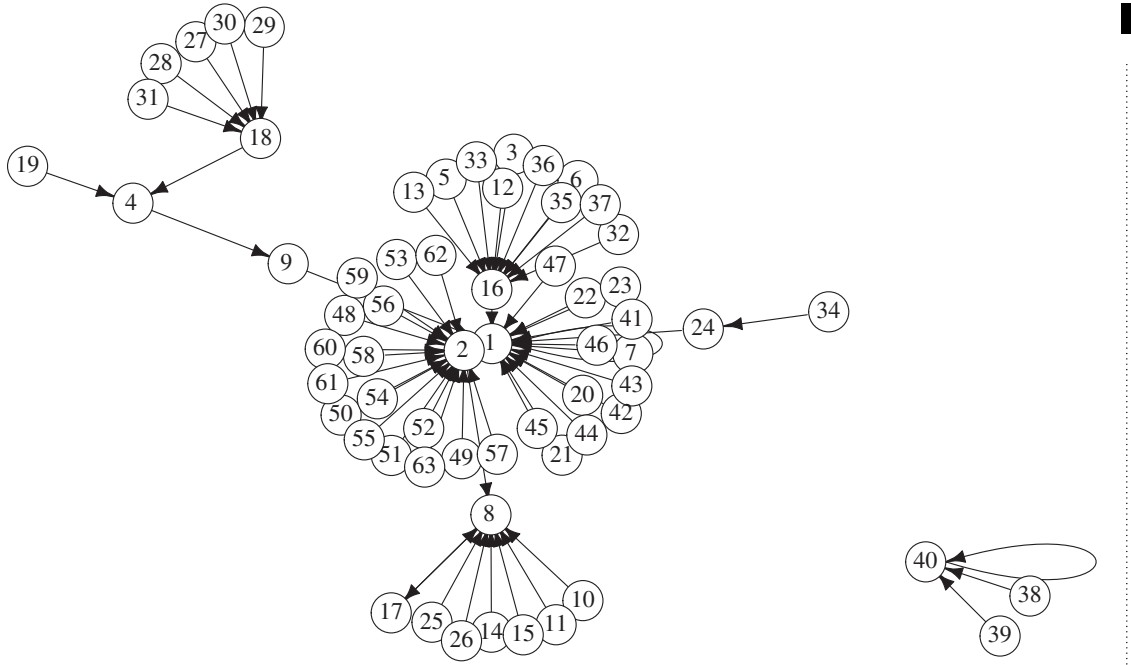

**Figure 12.** Graph of $g$ at $t = 41$.

filament [51]. Later, we realized that it might be unrealistic to expect someone to initiate and record travelling localizations (solitons, impulses) on a single actin filament. Therefore, we developed a numerical model of spikes propagating on a network of actin filament bundles and demonstrated that such a network can implement Boolean gates [33].

In the present paper, we reconsidered the whole idea of the information processing on actin networks and designed an actin droplet machine. The machine is a model of a three-dimensional network, based on an experimental network developed in a droplet, which executes mapping $F$ of a space of binary strings of length $k$ on itself. The machine acts as a finite state machine, which behaviour at a low level is governed by localizations travelling along the networks and interacting with each other. By focusing on a single element of a string, i.e. a single location of an electrode, we can reconstruct $k$ functions with $k$ arguments, as we have exemplified at the end of the §3. The exact structure of each $k$-ary function is determined by $F$, which, in turn, is determined by the exact architecture of a three-dimensional actin network and a configuration of electrodes.

Thus, potential future directions could be in detailed analysis of possible architectures of actin networks developed in laboratory experiments and evaluation on how far an exact configuration of electrodes affects the structure of mapping $F$ and corresponding distribution of functions implementable by the actin droplet machine. The ultimate goal would be to implement actin droplet machines in laboratory experiments and to cascade several machines into a multi-processors computing architecture.

Conventional hardware is static. Actin networks reconfigure dynamically: some filaments disappear by depolymerization, new filaments appear by polymerization. This is not a disadvantage of the actin network computers because: (1) they operate with the speed several orders more than actin treadmilling rate, (2) actin networks can be stabilized, (3) we can employ dynamic reconfigurablity in the computation.

A computation in actin bundle networks is implemented with travelling mechanical or electrical signals. Thus, we could estimate the speed of the signals propagation would be $10^6 \, \mu m \, s^{-1}$, for sound solitons, or $10^5$–$10^8 \, \mu m \, s^{-1}$, for action potential speed [52]. Let us take the lowest estimate $10^5 \, \mu m \, s^{-1}$. Assuming maximum linear size of an actin droplet machine is $ca$ 250 µm, the machine can process about 400 parallel inputs per second, thus operating at 0.4 kHz frequency. Commonly, actin polymerization speed is estimated to be $4 \times 10^{-1} \, \mu m \, s^{-1}$ [53]. An acting bundle has up to 500 actin filaments, which will not fail simultaneously. In fact, we have seen that the networks, once formed, could remain stable over hours without major rearrangements. In contrast with cells, no actin accessory proteins were used and no energy in the form of ATP was provided. The structures self-assembled solely driven by thermodynamic arguments into a stable, frozen state. If we would neglect these experimental findings and would assume an active network with high treadmilling rates, we

can consider the network being fixed for at least 10 s, which allows us to execute up to $4 \times 10^3$ cycles of computation.

The life-time of the fixed network can be even substantially changed by using accessory proteins such as purely synthetic actin crosslinkers from DNA and peptides [54], increasing a ratio of integrine [55] and drebrin [56] peptides in the matrix solution, hardening the filaments with $\alpha$-actinin [57] and stabilizing the filament with synthetic mini-nebuline [58,59]. Using accessory proteins such as gelsolin, cofilin, formin and myosins would even allow us to speed up potential reconfiguration effects, enabling us to build up a dynamic computing system [6].

Dynamical reconfiguration of actin network computers can be used as an advantage for accelerating Boolean satisfiability solvers [60], reconfigurable data flow machine for implementing atomic functional programming languages [61], dynamical genetic programming on evolvable Boolean networks [62,63], and cryptographic applications [64].

Data accessibility. Supporting data are available at [43].

Authors' contributions. F.H. and J.S. provided experimental laboratory data. A.A. designed the model. All authors analysed the results and co-wrote the paper.

Competing interests. The authors declare that they have no competing interests.

Funding. A.A. was partially supported by EPSRC grant no. EP/P016677/1.

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
