## [Reviewer comments · Royal Society Open Science]

Review History

RSOS-191135.R0 (Original submission)

Review form: Reviewer 1

Is the manuscript scientifically sound in its present form?

Yes

Are the interpretations and conclusions justified by the results?

Yes

Is the language acceptable?

Yes

Do you have any ethical concerns with this paper?

No

Have you any concerns about statistical analyses in this paper?

No

Recommendation?

Accept as is

Comments to the Author(s)

This is an excellent paper which discusses in practical terms how to implement a computational device based on actin networks. General ideas on the use of proteins and protein-based hybrid devices for computational applications have been discussed previously. This paper provides a practically implementable program to construct a device with inputs, gates and outputs on the basis of actin filament networks that send soliton-like signals along its length. This paper can open a new area of nanotechnology and hence should be published due to its potentially very high impact. The technical aspects are sound and the narrative is easy to follow.

Review form: Reviewer 2

Is the manuscript scientifically sound in its present form?

No

Are the interpretations and conclusions justified by the results?

No

Is the language acceptable?

Yes

Do you have any ethical concerns with this paper?

No

Have you any concerns about statistical analyses in this paper?

No

Recommendation?

Major revision is needed (please make suggestions in comments)

Comments to the Author(s)

The work is interesting and is likely to be eventually worthy of publication.

However, the current manuscript has several flaws, including clarity about what has been done and why. It also fails to provide sufficient context and discussion.

I would encourage the authors to persist, but the paper needs to improve.

Technically, there is a critical (to my understanding) mistake in the first unnumbered formula, in section 2. From the text, I would deduce that a circle should be a star in the second line.

There is another critical mistake in the third line of section 2.3: what is "+"?

The experiments in this paper appear to be entirely computational, but the actin network comes from a physical experiment (fig 2).

The physical network was mapped to the computational network according to the procedure in the last paragraph of section 2.1.

The choice of parameters for the mapping though seems arbitrary, and they would lead to different networks, which raises doubts about the significance of the Boolean function extracted in Section 4. Why would such a function be useful to compute something of interest?

And what does "compressed in the z-axis" mean?

Section 3: a complete mystery to me. I would suggest to just delete it because it presents some statistics that have no computational meaning, and it does not seem relevant to what follows. Anyway: What do you mean that you "counted the number of gates, OR etc."? How did you identify these gates? What are the inputs and the outputs?

The core of the paper picks 6 random points on the networks (the "electrodes") and interprets them as inputs and outputs for a Boolean function, and to trace the evolution of the related Boolean states over time. Simulations are carried out by cellular automata. The resulting Boolean function is presented at the end of section 4.

Discussion.

Although this is not stated very explicitly, it seems that the purpose of this paper is to show that an actin network can be used as a nano-device, i.e. not for any biological purpose or function. Note that the whole purpose (and hence the intended audience) of the paper is stated only in the last sentence of the abstract, and the last sentence of the Discussion section, and never convincingly argued.

The fact that actin networks can propagate excitations is definitely quite interesting, but is this purely accidental, or does it perform some biological function? How was the original network chosen? Was it "frozen" and photographed at a random time? As far as I know, actin networks are highly dynamic, continuously remodeling themselves even without the activity of other proteins. Or do you assume that the network remains fixed (and how?). And how could you ever use it if not?

In addition to the biological randomness, the electrodes are chose randomly: what effect has this choice on the properties of the network?

None of that is discussed. The paper comes out as a dry presentation of network statistics of a fixed and arbitrarily selected system, without any indication about the possible significance of these statistics or of the network itself.

Some of the necessary background might be contained in reference 4 and 6, in which case it should be included in this paper.

I might normally read some of the referenced articles, but unfortunately 4 and 6 are unpublished, and I cannot referee three papers for one.

Decision letter (RSOS-191135.R0)

05-Aug-2019

Dear Dr Adamatzky,

The editors assigned to your paper ("Actin droplet machine") have now received comments from reviewers. We would like you to revise your paper in accordance with the referee and Associate Editor suggestions which can be found below (not including confidential reports to the Editor). Please note this decision does not guarantee eventual acceptance.

Please submit a copy of your revised paper before 28-Aug-2019. Please note that the revision deadline will expire at 00.00am on this date. If we do not hear from you within this time then it will be assumed that the paper has been withdrawn. In exceptional circumstances, extensions may be possible if agreed with the Editorial Office in advance. We do not allow multiple rounds of revision so we urge you to make every effort to fully address all of the comments at this stage. If deemed necessary by the Editors, your manuscript will be sent back to one or more of the original reviewers for assessment. If the original reviewers are not available, we may invite new reviewers.

- Data accessibility

<http://datadryad.org/submit?journalID=RSOS&manu=RSOS-191135>

- Competing interests

- Authors' contributions

All submissions, other than those with a single author, must include an Authors' Contributions section which individually lists the specific contribution of each author. The list of Authors

should meet all of the following criteria; 1) substantial contributions to conception and design, or acquisition of data, or analysis and interpretation of data; 2) drafting the article or revising it critically for important intellectual content; and 3) final approval of the version to be published.

- Acknowledgements

- Funding statement

on behalf of Dr Francois Fages (Associate Editor) and Marta Kwiatkowska (Subject Editor)
openscience@royalsociety.org

Associate Editor's comments (Dr Francois Fages):

Comments to the Author:

Dear authors,

Your paper has been reviewed by two reviewers and it appears that a major revision is required to improve your communication and make a nice publication. You are invited to take into consideration all the criticisms of the second reviewer in preparing a revised version of your manuscript.

Best regards
F. Fages

Reviewers' Comments to Author:

Reviewer: 1

Comments to the Author(s)

This is an excellent paper which discusses in practical terms how to implement a computational device based on actin networks. General ideas on the use of proteins and protein-based hybrid devices for computational applications have been discussed previously. This paper provides a practically implementable program to construct a device with inputs, gates and outputs on the basis of actin filament networks that send soliton-like signals along its length. This paper can open a new area of nanotechnology and hence should be published due to its potentially very high impact. The technical aspects are sound and the narrative is easy to follow.

Reviewer: 2

Comments to the Author(s)

The work is interesting and is likely to be eventually worthy of publication. However, the current manuscript has several flaws, including clarity about what has been done and why. It also fails to provide sufficient context and discussion. I would encourage the authors to persist, but the paper needs to improve.

Technically, there is a critical (to my understanding) mistake in the first unnumbered formula, in section 2. From the text, I would deduce that a circle should be a star in the second line. There is another critical mistake in the third line of section 2.3: what is "+"?

The experiments in this paper appear to be entirely computational, but the actin network comes from a physical experiment (fig 2).

The physical network was mapped to the computational network according to the procedure in the last paragraph of section 2.1.

The choice of parameters for the mapping though seems arbitrary, and they would lead to different networks, which raises doubts about the significance of the Boolean function extracted in Section 4. Why would such a function be useful to compute something of interest?

And what does "compressed in the z-axis" mean?

Section 3: a complete mystery to me. I would suggest to just delete it because it presents some statistics that have no computational meaning, and it does not seem relevant to what follows. Anyway: What do you mean that you "counted the number of gates, OR etc."? How did you identify these gates? What are the inputs and the outputs?

The core of the paper picks 6 random points on the networks (the "electrodes") and interprets them as inputs and outputs for a Boolean function, and to trace the evolution of the related Boolean states over time. Simulations are carried out by cellular automata.

The resulting Boolean function is presented at the end of section 4.

Discussion.

Although this is not stated very explicitly, it seems that the purpose of this paper is to show that an actin network can be used as a nano-device, i.e. not for any biological purpose or function. Note that the whole purpose (and hence the intended audience) of the paper is stated only in the last sentence of the abstract, and the last sentence of the Discussion section, and never convincingly argued.

The fact that actin networks can propagate excitations is definitely quite interesting, but is this purely accidental, or does it perform some biological function?

How was the original network chosen? Was it "frozen" and photographed at a random time?

As far as I know, actin networks are highly dynamic, continuously remodeling themselves even without the activity of other proteins.

Or do you assume that the network remains fixed (and how?). And how could you ever use it if not?

In addition to the biological randomness, the electrodes are chose randomly: what effect has this choice on the properties of the network?

None of that is discussed. The paper comes out as a dry presentation of network statistics of a fixed and arbitrarily selected system, without any indication about the possible significance of these statistics or of the network itself.

Some of the necessary background might be contained in reference 4 and 6, in which case it should be included in this paper.

I might normally read some of the referenced articles, but unfortunately 4 and 6 are unpublished, and I cannot referee three papers for one.

Author's Response to Decision Letter for (RSOS-191135.R0)

See Appendix A.

RSOS-191135.R1 (Revision)

Review form: Reviewer 2

Is the manuscript scientifically sound in its present form?

Yes

Are the interpretations and conclusions justified by the results?

Yes

Is the language acceptable?

Yes

Do you have any ethical concerns with this paper?

No

Have you any concerns about statistical analyses in this paper?

No

Recommendation?

Accept as is

Comments to the Author(s)

The authors have made a good effort of answering my questions, which were largely due to lack of sufficient explanations and details. With the expanded Introduction and Conclusions the paper is easier to read and its purpose and constraints are much clearer. Technical details have also been expanded, clarifying some points. I believe it is now acceptable for publication.

Decision letter (RSOS-191135.R1)

04-Nov-2019

Dear Dr Adamatzky,

I am pleased to inform you that your manuscript entitled "Actin droplet machine" is now accepted for publication in Royal Society Open Science.

Kind regards,

Andrew Dunn

Senior Publishing Editor
Royal Society Open Science
openscience@royalsociety.org

on behalf of Dr Francois Fages (Associate Editor) and Marta Kwiatkowska (Subject Editor)
openscience@royalsociety.org

Associate Editor Comments to Author (Dr Francois Fages):

Associate Editor: 1

Comments to the Author:

Dear Authors

It is my pleasure to inform you that your revision has fully clarified the few remaining points and that your paper is now accepted as is.

Best regards

Reviewer comments to Author:

Reviewer: 2

Comments to the Author(s)

The authors have made a good effort of answering my questions, which were largely due to lack of sufficient explanations and details. With the expanded Introduction and Conclusions the paper is easier to read and its purpose and constraints are much clearer. Technical details have also been expanded, clarifying some points. I believe it is now acceptable for publication.

Appendix A

Response to Reviewers: Actin droplet machine

Andrew Adamatzky, Florian Huber, Jörg Schnauß

The authors are grateful to the reviewers for their positive and constructive comments. One reviewer characterised the paper as excellent with easy to follow narrative. Another reviewer raised questions about few technical details that have been addressed in the revised version and are listed below.

Reviewer: Technically, there is a critical (to my understanding) mistake in the first unnumbered formula, in section 2. From the text, I would deduce that a circle should be a star in the second line. There is another critical mistake in the third line of section 2.3: what is ”+”?

Authors: Thank you very much for spotting this misprint, that should indeed have been \star and was updated. The mentioned ‘+’ should be \star . Both instances have been corrected in the current version of the manuscript.

Reviewer: The choice of parameters for the mapping though seems arbitrary, and they would lead to different networks, which raises doubts about the significance of the Boolean function extracted in Section 4. Why would such a function be useful to compute something of interest? And what does ”compressed in the z-axis” mean?

Authors: The choice is not arbitrary but quite conventional from an image processing point of view when a 3D object is extracted from the slices. The RGB thresholds of pixels in the slices are chosen to discriminate an object, in our case actin bundle network, from the background. ‘Compression’ means that the actual physical thickness of each slide was not fully taken into account for the arrangement of voxels. In our model, we treat each voxel to have identical dimensions along x, y, and z. This represents an underestimation of the actual extension along the z-axis, hence effectively acting as a compression. As the voxel automaton neighbourhood radius was chosen to be $r = 3$, the ‘compression’ did not affect quality of the excitation wave-fronts yet allowed for a feasible speed of numerical modelling.

Reviewer: Section 3: a complete mystery to me. I would suggest to just delete it because it presents some statistics that have no computational meaning, and it does not seem relevant to what follows. Anyway: What do you mean that you ”counted the number of gates, OR etc.”? How did you identify these gates? What are the inputs and the outputs?

Authors: We agree that in the original version of the paper the purpose of the section was not entirely clear, therefore we updated the text. We made the title more concrete

Selecting excitation threshold and refractory delay to maximise a number of logical gates

to show that we analysed distributions of two-input-one-output logical functions to select parameters for maximising the number of the functions realised.

We have also added a remark to better motivate the purpose of the section:

To design an actin droplet machine with complex behaviour we need to find values of refractory delay and excitation threshold for which the actin bundles network executes a maximum of Boolean gates.

and also highlighted this at the end of the section:

The gate frequency analysis presented in this section, allow us to choose $\theta = 7$ and $\delta = 20$ for an actin droplet machine constructed in the next section.

With regards to a procedure of Boolean gates detection we provided more explanations:

We stimulated the network with all possible configurations of inputs, recorded the network's electrical dynamics and then extracted logical gates as follows. For each possible combination (i, j, k) , $1 \leq i, j, k \leq 6$, $i \neq j$, $i \neq k$, $j \neq k$, we considered electrodes \mathcal{E}_i and \mathcal{E}_j to be inputs, representing Boolean variables x and y , respectively, and electrode \mathcal{E}_k as output electrode, representing result of a Boolean function. To input $x = \text{TRUE}$ we applied a current to electrode \mathcal{E}_i , to input $y = \text{TRUE}$ to electrode \mathcal{E}_j . Then we record the potential at electrode \mathcal{E}_k . Two-input-one-output logical functions were extracted from the spiking events as follows. Assume each spike represents logical TRUE and that spikes being less than six iterations closer to each other happen at the same moment. Then a representation of gates by spikes and their combination will be as shown in Fig. 1.

Reviewer: Although this is not stated very explicitly, it seems that the purpose of this paper is to show that an actin network can be used as a nano-device, i.e. not for any biological purpose or function. Note that the whole purpose (and hence the intended audience) of the paper is stated only in the last sentence of the abstract, and the last sentence of the Discussion section, and never convincingly argued.

Authors: To add more stress on the ultimate goal of designing computers from actin networks we added the following text in the Introduction section:

As actin networks can implement logical functions, they can compute. So, in [2] we proposed a road map to experimental implementation of cytoskeleton-based computing devices. We proposed that collision-based cytoskeleton computers implement logical gates via interactions between travelling localisation: voltage solitons on actin filaments or tubulin microtubules bundles. An architecture of cytoskeleton computers can be developed via programmable polymerisation of actin networks. Such cytoskeleton computers would take data

Figure 1: Representation of two-inputs-one-output Boolean gates by combinations of spikes. Black dotted line shows the potential at an output electrode when the network was stimulated by input pair $(x, y) = (\text{FALSE}, \text{TRUE})$, red solid by $(\text{TRUE}, \text{FALSE})$ and green dashed by $(x, y) = (\text{TRUE}, \text{TRUE})$.

via electrical and optical means, the signals (solitons, conformational defects) initiated by the input stimuli will be travelling along the network and the computation will be implemented via collisions of the signals at the structural gates of the network.

We have also provided more explanations with regards to what has been done previously:

In a previous paper [1] we demonstrated, using numerical integration of FitzHugh-Nagumo model, that a two-dimensional actin network realised k -ary Boolean functions $G : \{0, 1\}^k \rightarrow \{0, 1\}$, when k input electrodes and one output electrodes are employed.

Reviewer: The fact that actin networks can propagate excitations is definitely quite interesting, but is this purely accidental, or does it perform some biological function?

Authors: To make it more clear to readers that the paper does not deal with any biological meanings we added the following statement in the Introduction:

Results presented in the paper give a rather ‘computer engineering’ view on a computation implementable with travelling localisations on acting bundle networks. We do not speculate about potential biological meanings of the phenomena described. That could be a scope of future studies.

Reviewer: How was the original network chosen? Was it “frozen” and photographed at a random time? As far as I know, actin networks are highly dynamic, continuously remodeling themselves even without the activity of other proteins. Or do you assume that the network remains fixed (and how?). And how could you ever use it if not?

Authors: This is an interesting point, which could be even used in future studies to make this new technological approach even more versatile. Generally, the computation is much faster than the polymerisation rates of actin filaments. Additionally, actin networks in the tested configuration were experimentally observed to remain stable for hours since they are not based on single filaments but bundled actin structures. This arrangement of filaments is very stable representing a frozen state without visible structural rearrangements. In our case, no actin accessory proteins were used, which are commonly employed by cells to speed up the actin polymerisation process. Additionally, no energy in form of ATP is consumed and the system settles purely thermodynamically driven into an energetic minimum. A cell is always far from thermodynamic equilibrium consuming a high amount of energy to increase the turnover dynamics of actin networks. In addition to this intriguing property of actin to self-assemble into these stable structures, it could even be possible to initiate changes in a controlled fashion by adding accessory proteins. This, however, is beyond the scope of this study. We now address these issues in the Discussion section as follows:

A computation in actin bundle networks is implemented with travelling mechanical or electrical signals. Thus, we could estimate a speed of the signals propagation would be $10^6 \mu\text{m/s}$, for sound solitons, or $10^5\text{--}10^8 \mu\text{m/s}$, for action potential speed [14]. Let us take the lowest estimate $10^5 \mu\text{m/s}$. Assuming maximum linear size of an actin droplet machine is ca. $250 \mu\text{m}$, the machine can process about 400 parallel inputs per second, thus operates at 0.4 kHz frequency. Commonly, actin polymerisation speed is estimated to be $4 \cdot 10^{-1} \mu\text{m/s}$ [12]. An acting bundle has up to 500 actin filaments, which could not fail simultaneously. In fact, we have seen that the networks, once formed, could remain stable over hours without major rearrangements. In contrast to cells, no actin accessory proteins were used and no energy in form of ATP was provided. The structures self-assembled solely driven by thermodynamic arguments into a stable, frozen state. If we would neglect these experimental findings and would assume an active network with high treadmilling rates, we can consider the network being fixed for at least 10 s, which allows us to execute up to $4 \cdot 10^3$ cycles of computation.

The life-time of the fixed network can be even substantially changed by using accessory proteins such as purely synthetic actin crosslinkers from DNA and peptides [8], increasing a ratio of integrin [9] and drebrin [10] peptides in the matrix solution, hardening the filaments with α -actinin [15] and stabilising the filament with synthetic mini-nebulin [11, 5]. Using accessory proteins such as gelsolin, cofilin, formin and myosins would even allow to speed up potential reconfiguration effects enabling to build up a dynamic computing system [7].

Dynamical reconfiguration of actin network computers can be used as an advantage for accelerating Boolean satisfiability solvers [16], reconfigurable data flow machine for implementing atomic functional programming languages [6], dynamical genetic programming on evolvable Boolean networks [3, 4], cryptographic applications [13].

Reviewer: In addition to the biological randomness, the electrodes are chosen randomly: what effect has this choice on the properties of the network?

Authors: The network is not entirely random, the actin bundles connecting nucleation sites are affine to edges in the 3D Delaunay triangulation. An exact distribution of the nucleation sites might require additional analysis in future works, it appears that each site is positioned at nearly the same distance from its geographic neighbours. Thus we can hypothesize that the nucleation sites make a uniform tiling of the droplet space. With regards to the position of electrodes, indeed, the global state transition graphs might be determined by the configurations of electrodes. Future studies will be required to understand how a topology of the global state transition graphs is affected by the electrodes configurations.

References

- [1] Andrew Adamatzky, Florian Huber, and Jörg Schnauß. Computing on actin filament bundles. *arXiv preprint arXiv:1903.10186*, 2019.
- [2] Andrew Adamatzky, Jack Tuszynski, Joerg Pieper, Dan V Nicolau, Rossalia Rinaldi, Georgios Sirakoulis, Victor Erokhin, Joerg Schnauss, and David M Smith. Towards cytoskeleton computers. A proposal. In Andrew Adamatzky, Selim Akl, and Georgios Sirakoulis, editors, *From parallel to emergent computing*. CRC Group/Taylor & Francis, 2019.
- [3] Larry Bull. On dynamical genetic programming: Simple boolean networks in learning classifier systems. *International Journal of Parallel, Emergent and Distributed Systems*, 24(5):421–442, 2009.
- [4] Larry Bull. On the evolution of boolean networks for computation: a guide rna mechanism. *International Journal of Parallel, Emergent and Distributed Systems*, 31(2):101–113, 2016.
- [5] Miensheng Chu, Carol C Gregorio, and Christopher T Pappas. Nebulin, a multi-functional giant. *Journal of Experimental Biology*, 219(2):146–152, 2016.
- [6] Christophe Giraud-Carrier. A reconfigurable data flow machine for implementing functional programming languages. *Sigplan Notices*, 29(9):22–28, 1994.
- [7] Florian Huber, Jörg Schnauß, Susanne Rönicke, Philipp Rauch, Karla Müller, Claus Fütterer, and Josef A. Käs. Emergent complexity of the cytoskeleton: from single filaments to tissue. *Advances in Physics*, 62(1):1–112, 2013.
- [8] Jessica S Lorenz, Jörg Schnauß, Martin Glaser, Martin Sajfutdinow, Carsten Schuldt, Josef A Käs, and David M Smith. Synthetic transient crosslinks program the mechanics of soft, biopolymer-based materials. *Advanced Materials*, 30(13):1706092, 2018.
- [9] Andrew B McGeachie, Lorenzo A Cingolani, and Yukiko Goda. A stabilising influence: integrins in regulation of synaptic plasticity. *Neuroscience research*, 70(1):24–29, 2011.
- [10] Mouna A Mikati, Elena E Grintsevich, and Emil Reisler. Drebrin-induced stabilization of actin filaments. *Journal of Biological Chemistry*, 288(27):19926–19938, 2013.

- [11] Christopher T Pappas, Paul A Krieg, and Carol C Gregorio. Nebulin regulates actin filament lengths by a stabilization mechanism. *The Journal of cell biology*, 189(5):859–870, 2010.
- [12] Thomas D Pollard and Mark S Mooseker. Direct measurement of actin polymerization rate constants by electron microscopy of actin filaments nucleated by isolated microvillus cores. *The Journal of cell biology*, 88(3):654–659, 1981.
- [13] Francisco Rodríguez-Henríquez, Nazar Abbas Saqib, Arturo Díaz Pérez, and Cetin Kaya Koc. *Cryptographic algorithms on reconfigurable hardware*. Springer Science & Business Media, 2007.
- [14] Knut Schmidt-Nielsen. *Animal physiology: adaptation and environment*. Cambridge University Press, 1997.
- [15] Jingyuan Xu, Yiider Tseng, and Denis Wirtz. Strain hardening of actin filament networks regulation by the dynamic cross-linking protein α -actinin. *Journal of Biological Chemistry*, 275(46):35886–35892, 2000.
- [16] Peixin Zhong, Margaret Martonosi, Pranav Ashar, and Sharad Malik. Solving boolean satisfiability with dynamic hardware configurations. In *International Workshop on Field Programmable Logic and Applications*, pages 326–335. Springer, 1998.